# New Photomagnetic Ionic Salts Based on [Mo^IV(CN)_8]^4- and [W^IV(CN)_8]^4- Anions †

Xinghui Qi [1], Philippe Guionneau [1,*], Enzo Lafon [1], Solène Perot [1], Brice Kauffmann [2] and Corine Mathonière [1,3,*]

1 Université de Bordeaux, CNRS, Bordeaux INP, ICMCB, UMR 5026, F-33600 Pessac, France; xinghui.qi@icmcb.cnrs.fr (X.Q.); enzo.lafon@etu.u-bordeaux.fr (E.L.); solene.perot@etu.u-bordeaux.fr (S.P.)

2 Université de Bordeaux, IECB, UMS 3033, Institut Européen de Chimie et Biologie, 2 rue Escarpit, F-33600 Pessac, France; b.kauffmann@iecb.u-bordeaux.fr

3 Université de Bordeaux, CNRS, Centre de Recherche Paul Pascal, UMR 5031, F-33600 Pessac, France

* Correspondence: philippe.guionneau@icmcb.cnrs.fr (P.G.); corine.mathoniere@u-bordeaux.fr (C.M.)

† Dedicated to Professor Peter Day, this article belongs to the Special Issue—Perspectives on Molecular Materials—A Tribute to Professor Peter Day.

**Abstract:** Three new ionic salts containing [M(CN)_8]^4- (M = Mo^IV and W^IV) were prepared using large complex cations based on a non-conventional motif built with the tris(2-aminoethyl)amine (noted hereafter tren) ligand, [{M′(tren)}_3(μ-tren)]^6+ (M′ = Cu^II and Zn^II). The crystal structures of the three compounds show that the atomic arrangement is formed by relatively isolated anionic and cationic entities. The three compounds were irradiated with a blue light at low temperature, and show a significant photomagnetic effect. The remarkable properties of these compounds are (i) the long-lived photomagnetic metastable states for the [Mo(CN)_8]^4--based compounds well above 200 K and (ii) the rare efficient photomagnetic properties of the [W(CN)_8]^4--based compound. These photomagnetic properties are compared with the singlet-triplet conversion recently reported for the K_4[Mo(CN)_8]·2H_2O compound.

**Keywords:** coordination compounds; octacyanometalates; photomagnetism

## 1. Introduction

The richness of the photochemistry of K_4[MoCN_8]·2H_2O in solution has been known for a few decades [1,2]. Irradiation of aqueous solutions of [MoCN_8]^4- in its ligand field bands (350–400 nm) allows photosubstitution reactions, with the isolation of different species, [MoCN_7(H_2O)]^3- or [MoCN_7(OH)]^4- [3], that depend on the pH of the solutions. The [Mo(CN)_8]^4- complex can be used as a building block to form a prolific series of both polynuclear compounds and coordination polymers [4,5]. Several of these compounds show interesting photomagnetic properties based on either photo-induced electron transfer for systems exhibiting Metal-to-Metal Charge Transfer (MMCT) or Singlet-Triplet formation on Mo^IV [5].

More recently, an intriguing breakage of Mo–CN bond has been discovered in the crystalline solid state after blue light irradiation at 10 K for the K_4[MoCN_8]·2H_2O complex [6]. The removal of one CN ligand from the Mo coordination sphere is accompanied by the capture of the free CN group in the crystal lattice by water molecules. Interestingly, first, the photo-induced effect is accompanied by a spin change, and second, it is fully reversible through a thermal heating. This phenomenon of decoordination/coordination is quite common in solution, in particular for Ru [7,8], Fe [9] and Ni [10] complexes, but it remains rare in the solid state and even rarer in the crystalline solid state without loss of crystallinity. So far, only very few compounds show a reversible bond breaking. One example is based on a Co complex [11] that exhibits a dynamic bond switching with a modulation of the ligand field and the orbital momentum of the metal ion. The second

example is shown in a family of spin crossover Fe$^{II}$ [12,13] complexes where the decoordination/coordination process can be thermally and photo-induced, as it is the case in the K$_4$[Mo(CN)$_8$]·2H$_2$O complex [6]. To explore if this phenomenon can be extended to other Mo/W-based systems, we have started a systematic study of compounds having non-bridged [M(CN)$_8$]$^{4-}$ complexes. In this work, we will present the synthesis, the structural and magnetic characterizations of two new anionic [Mo(CN)$_8$]$^{4-}$ complexes crystallized with large coordination cations containing Cu$^{2+}$ and Zn$^{2+}$ ions. Additionally, we extend this study to the analogous [W(CN)$_8$]$^{4-}$ complex. For these three new systems, significant photomagnetic responses have been obtained. These properties have been analyzed with a model based on the recent report of the photo-induced single-triplet crossover [6].

## 2. Results and Discussion

### 2.1. Synthesis and Characterization

The compounds **1** (for [{Cu(tren)}$_3$(μ-tren)]$_4$[Mo(CN)$_8$]$_6$·45H$_2$O·2CH$_3$OH), **2** (for [{Zn(tren)}$_3$(μ-tren)]$_2$[Mo(CN)$_8$]$_3$·18H$_2$O) and **3** (for [{Zn(tren)}$_3$(μ-tren)]$_2$[W(CN)$_8$]$_3$·17H$_2$O) were obtained by mixing solutions containing 3d divalent metal M$^{2+}$ cations, tren ligand and [M(CN)$_8$]$^{4-}$ anions following two different methods. Green crystals of compound **1** were prepared by a layering method by the diffusion of an aqueous solution of CuCl$_2$·2H$_2$O and tren ligand into the solution of K$_4$[Mo$^{IV}$(CN)$_8$]·2H$_2$O, the two solutions being separated by a layer of methanol and water. Compounds **2** and **3** were prepared by one-pot method. By mixing solutions of ZnCl$_2$, tren ligand and K$_4$[M$^{IV}$(CN)$_8$]·2H$_2$O (M = Mo (**2**) and W (**3**)), a clear yellow solution was obtained. Then the slow addition of about 1.5 mL of methanol led to the formation of target yellow crystals of **2** and **3** after one night.

Infrared spectra (IR) for compounds **1**, **2** and **3** are very similar (Figure S1 and Tables S1 and S2). The three compounds show the characteristics bands of the tren ligand at 1600, 1450, 1300, and 995 cm$^{-1}$. **1** and **2** show a broad band centered at 2098 cm$^{-1}$, the signature of terminal CN ligands coordinated to the Mo$^{4+}$ by the C atoms. For compound **3**, this band appears at 2091 cm$^{-1}$ in agreement with the coordination of the terminal CN to the W$^{4+}$ by the C atoms. These spectra indicate that **1**, **2** and **3** are ionic salts and that the chemical environment of the [M(CN)$_8$]$^{4-}$ anions are similar in the three compounds.

UV-Vis spectra of compounds **2** and **3** are similar to that of K$_4$[Mo(CN)$_8$]·2H$_2$O and K$_4$[W(CN)$_8$]·2H$_2$O [14], while there is a broad absorption with maximum and shoulders at 877 and 657 nm for compound **1** (Figure S2 and Table S3). These bands also appear in [Cu(tren)]$^{2+}$ complexes in square pyramidal geometry [15], and suggest that the tren ligand acts in **1** as a tetradentate ligand for the Cu$^{2+}$ ion. This means that a fifth ligand is necessary to assure for the Cu$^{2+}$ ion a bipyramid geometry. It is worth noteing that for **1** no additional transition except the transitions observed in its precursors is observed, suggesting the absence of an outer-sphere charge transfer in Cu$^{2+}$/Mo$^{4+}$ pairs in **1**.

### 2.2. Crystal Structure Description of the Compounds

#### 2.2.1. [{Cu(tren)}$_3$(μ-tren)]$_4$[Mo(CN)$_8$]$_6$·45H$_2$O·2CH$_3$OH (**1**)

Single Crystal X-ray diffraction (SCXRD) analysis shows that **1** crystallizes in the monoclinic space group *P*2$_1$/*c*. As shown by the crystallographic data in Table A1, the unit cell of **1** is very large, leading to up to 323 atoms (without H atoms) with 3D coordinates in the asymmetric unit which certainly puts this compound in the category of the giant unit-cell ones (V > 20,000 Å$^3$). This makes the SCXRD crystal structure determination a challenge in itself. **1** is constructed by the assembly of [{Cu(tren)}$_3$(μ-tren)]$^{6+}$ cations and [Mo(CN)$_8$]$^{4-}$ anions with no covalent bond between them. The packing diagram of **1** is shown on Figure S4. There are six [Mo(CN)$_8$]$^{4-}$ anions and four [{Cu(tren)}$_3$(μ-tren)]$^{6+}$ cations in the asymmetric unit (Figure 1). The very large number of solvent molecules in the asymmetric unit (>40 water and/or methanol molecules) and the flexible arms for the [{Cu(tren)}$_3$(μ-tren)]$^{6+}$ cations cause structural disorder, which increases the difficulty in the crystal structure determination. Nevertheless, it is important to note that the crystal structure has been solved without ambiguity and confirmed by the diffraction

data investigation on several crystals from different batches. As a result, while the solvent part cannot be discussed in detail, the structural parameters and notably the 3D atomic coordinates for the anions and cations in **1** are, on the contrary, robust and can be discussed further. The crystallographic data and selected bond lengths and angles are presented in Table A1, Tables S4 and S7.

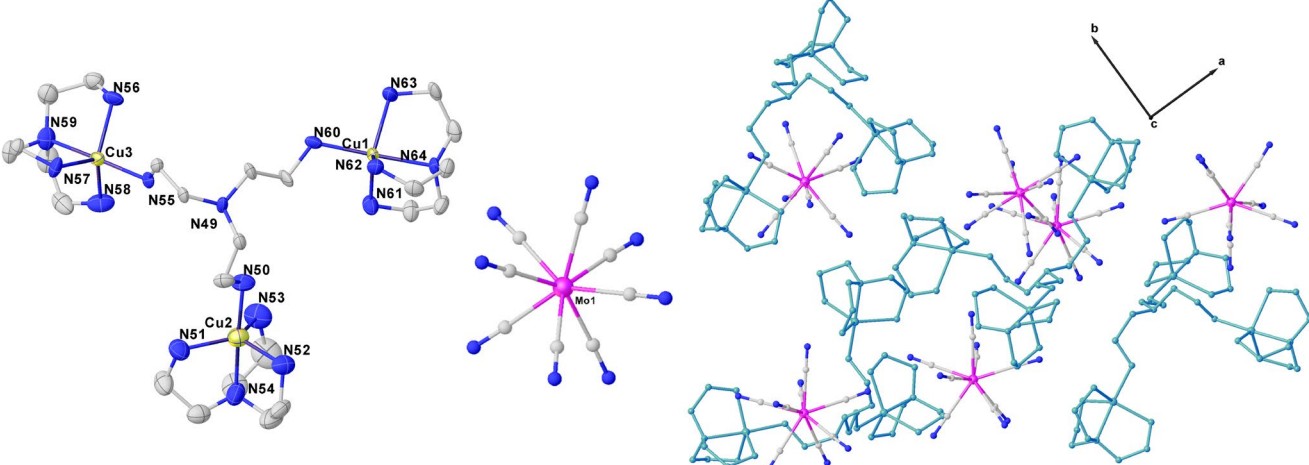

**Figure 1.** [{Cu(tren)}$_3$(μ-tren)]$^{6+}$ (**left**), [Mo(CN)$_8$]$^{4-}$ (**middle**) selected fragments with partial atoms labelling and asymmetric unit of **1** (without solvent entities) viewed along *c* with [{Cu(tren)}$_3$(μ-tren)]$^{6+}$ units in green (**right**). Color codes: N, blue; C, grey; Cu, yellow; Mo, pink.

The [{Cu(tren)}$_3$(μ-tren)]$^{6+}$ is constructed by one μ-tren linked to three Cu sites, where each copper site is blocked by another tren ligand. Therefore, the tren ligands serve as a tetradentate ligand for the copper sites, and as a tridentate ligand to link the three different Cu sites contained in the trimetallic cation. To the best of our knowledge, this unusual trinuclear copper complex cation has been reported only in the crystal structure of [Cu$_3$(tren)$_4$][Pt(CN)$_4$]$_3$·2H$_2$O [16]. As indicated by the continuous shape measurement (CShM) [17] values, all the copper sites adopt triangular bipyramidal geometry, with the exception of Cu9 site which corresponds to distorted square pyramidal geometry, in agreement with the UV-Visible spectra (see Table A2). For instance, the CShM value for Cu1 site is 0.420, corresponding to the triangular bipyramidal geometry and for Cu9 site is 1.491, corresponding to the square pyramidal geometry. The three copper sites in [{Cu(tren)}$_3$(μ-tren)]$^{6+}$ are arranged in the form of an irregular triangle, for example, with rather long Cu...Cu distances, such as as 7.450, 7.542, 9.156 Å for Cu1...Cu2, Cu2...Cu3 and Cu1...Cu3 distances, respectively.

The [Mo(CN)$_8$]$^{4-}$ anion is stabilized by wide-numerous N-H ... N≡C and O-H ... N≡C hydrogen bonds formed by the interaction of [{Cu(tren)$_3$(μ-tren)]$^{6+}$ or water molecules with [Mo(CN)$_8$]$^{4-}$ units, respectively. The selected bond lengths and angles for Mo sites are presented in Table S4. Average bond distances of Mo-C and C≡N are 2.179(10)/2.170(10)/2.176(14)/2.177(10)/2.153(13)/2.169(10) and 1.145(14)/1.141(14)/-1.135(6)/1.137(13)/1.157(10)/1.138(14) Å, respectively, while the average Mo-C≡N bond angles equal to 177.0(10)/176.7(10)/175.9(12)/177.5(9)/176.8(13)/177.3(10)°. All the Mo sites reveal a geometry close to the square antiprism (SAPR), as evidenced by continuous shape measurement (CShM) analysis (Table A2). The minimum Mo ... Mo distances are 9.65, 9.65, 9.77, 9.67, 9.80, 9.67 Å, which are much longer than the distance of 7.53 Å found in the reference compound K$_4$[Mo$^{IV}$(CN)$_8$]·2H$_2$O [6], but appear comparable to the values observed for [Ni(bipy)$_3$]$_2$[Mo(CN)$_8$]·12H$_2$O [18].

### 2.2.2. [{Zn(tren)}$_3$(μ-tren)]$_2$[Mo(CN)$_8$]$_3$·18H$_2$O (**2**)

Similarly to **1**, compound **2** is based on the blocks [{M′(tren)}$_3$(μ-tren)]$^{6+}$ (M′ = Cu (**1**) and Zn (**2**)) and [Mo(CN)$_8$]$^{4-}$ (Figure 2), but it crystallizes in the non-centrosymmetric

space group *Cc*. The unit-cell is about twice smaller than for **1**, leading to half of the asymmetric content, namely, three $[Mo(CN)_8]^{4-}$ anions, two $[\{Zn(tren)\}_3(\mu\text{-tren})]^{6+}$ and roughly half of the solvent (water) molecules (one formula unit). The crystal structure refinement is consequently of better quality than for **1**, reaching almost for **2** the standard criterion expected for small molecules, although here also the number of atoms in the asymmetric unit remains impressive (157 without H atoms). The crystallographic data and selected bond lengths and angles are presented in Table A1, Tables S5 and S8. In the same way as for **1**, the crystal structure of **2** contains a large number of solvent molecules that are difficult to localize, although here the determination of H atoms position could reasonably be conducted. However, it is hazardous to discuss in detail features that concern the solvent entities. The coordination metal ions are on the contrary very well defined and can be further discussed. Similar to $[\{Cu(tren)\}_3(\mu\text{-tren})]^{6+}$ found in **1**, the zinc sites adopt the triangular bipyramidal geometry, for example, with continuous shape measurement (CShM) value of 0.817 for Zn1 site (see Table A2). The coordination geometry of the Zn ions is close to the ones found in the reported trinuclear compound $MoZn_2$–tren [19]. The selected bond lengths and angles for Mo sites are presented in Table S5. Average bond distances of Mo-C and C≡N are 2.174(13)/2.166(12)/2.179(12) and 1.139(12)/1.150(12)/1.132(12) Å, respectively. The average Mo-C≡N bond angles are equal to 177.0(12)/177.7(15)/176.6(13)°. The shortest Mo . . . Mo distances are 9.48/9.46/9.46 Å, which fall in the same range as the ones found in compound **1**. The three Mo sites in **2** are also close to a square antiprism (SAPR) geometry as evidenced by the continuous shape measurement (CShM) values (Table A2).

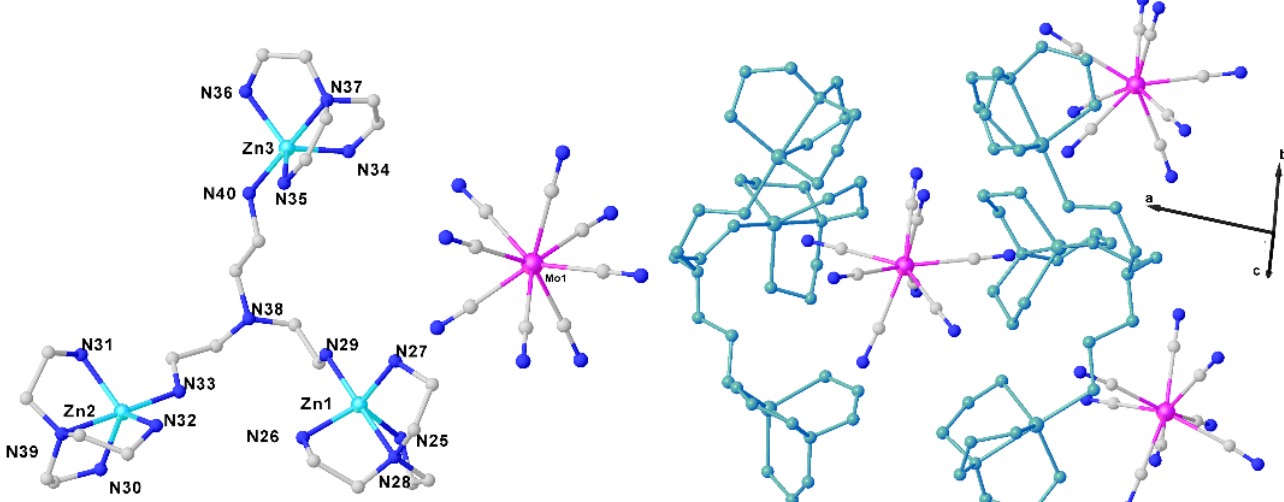

**Figure 2.** Selected $[\{Zn(tren)\}_3(\mu\text{-tren})]^{6+}$ (**left**), $[Mo(CN)_8]^{4-}$ (**middle**) fragments of **2** and asymmetric unit of **2** (without solvent entities) with $[\{Zn(tren)\}_3(\mu\text{-tren})]^{6+}$ units in green (**right**). Color codes: N, blue; C, grey; Zn, light blue; Mo, pink.

### 2.2.3. $[\{Zn(tren)\}_3(\mu\text{-tren})]_2[W(CN)_8]_3\cdot17H_2O$ (**3**)

The crystallographic data of **3** are given in Table A1. It adopts the same unit-cell and space group as **2** and atomic positions are very close in these two structures. The crystal structure of **3** is therefore similar to the one of **2** with the replacement of $[Mo(CN)_8]^{4-}$ anions by $[W(CN)_8]^{4-}$ anions. Crystal structure criteria of **3** are even slightly better than for **2**. The asymmetric unit of **3** is given in Figure S3, whereas the selected bond lengths and angles are reported in Tables S6 and S9. The $[W(CN)_8]^{4-}$ anions also adopt the SAPR geometry with CShM values comparable with those of the Mo sites found in **2**. The shortest W . . . W distances are 9.40/9.45/9.40 Å.

Notably, the crystal packing for **3** is quite different from that for **1** (Figure S4). Compounds **1** and **3** exhibit as two-dimensional and three-dimensional coordination polymers if considering the semi-coordination bonds, respectively.

The experimental powder X-ray diffraction (PXRD) patterns are globally consistent with the above SCXRD results but the large unit-cells combined with the low symmetry involved make the PXRD almost mute in term of reliable information (Figure S5). We can only note that the powders of **1** and **3** look more poorly crystalline than the powder of **2**.

*2.3. Magnetic and Photomagnetic Properties*

The magnetic and photomagnetic properties of **1**, **2** and **3** have been studied with microcrystalline powders sealed in a small PVE bag (see experimental section). Irradiations at 405 nm were selected because this wavelength fits with the energy range of one ligand field transition of the anions. For the three compounds, the magnetic properties (i.e., magnetization versus field at low temperatures and/or $\chi T$ versus temperature, $\chi$ being the magnetic susceptibility and $T$ the temperature) were first studied in the dark (curves named dark). At 10 K, the samples are irradiated and the time dependence of the magnetic properties is followed during light irradiation. After the light excitations, magnetizations versus field at low temperature are measured. Then the samples are heated again to evaluate the persistence of the photo-induced changes from 2 K to 300 K (curves named after blue irradiation). Finally, the compounds are measured again in the dark from 300 K to 10 K to check the reversibility of the photo-induced magnetic changes (curves named relaxation). All the magnetic curves shown below are normalized per $Cu_2Mo$ for **1**, $Zn_2Mo$ for **2** and $Zn_2W$ for **3**. This normalization will allow an easier comparison for the discussion of the results.

2.3.1. CutrenMo Compound (**1**)

As observed from the temperature dependence of the $\chi T$ and low-temperature magnetizations in the dark (Figure 3), **1** reveals a paramagnetic behavior with a $\chi T$ product equal to 0.80 cm$^3$ mol$^{-1}$ K in agreement with two $Cu^{2+}(3d^9)$ ions of $S = 1/2$ (per $Cu_2Mo$ units) with a Zeeman factor of $g = 2$ (Figure S6) and one diamagnetic $Mo^{4+}(4d^2$ in square antiprism geometry) ion. The superposition of the reduced magnetizations measured at 1.8, 3 and 5 K suggests the absence of magnetic anisotropy. When **1** is irradiated with a light of 405 nm, the value of $\chi T$ at 10 K increases from 0.79 to 1.4 cm$^3$ mol$^{-1}$ K after 25 h of irradiation (inset, Figure 3). Then the light is switched off, and low-temperature magnetizations at 1.8, 3 and 5 K were measured. The saturation magnetization at 1.8 K is now 3.09 Nβ, significantly higher than the value of 2 Nβ found before irradiation. The non-superposed reduced magnetizations suggest anisotropy in the photo-induced state. This observation is clearly different in the ground state. The temperature dependence of the $\chi T$ from 2 K to 300 K first increases to reach a plateau at 1.67 cm$^3$ mol$^{-1}$ K at 30 K. When compared with the $\chi T$ product before irradiation, the $\chi T$ value has increased with a maximum of 0.89 cm$^3$ mol$^{-1}$ K. Then, the $\chi T$ product slightly decreases monotonously to reach 1.43 cm$^3$ mol$^{-1}$ K at 250 K. Above this temperature, a faster decrease of the $\chi T$ product is observed. At 300 K, the $\chi T$ value is back to the value obtained before the light irradiation. A new plot $\chi T$ vs. $T$ (red curve on Figure 3) shows that the photo-induced process is reversible.

2.3.2. ZntrenMo Compound (**2**)

As **2** contains only diamagnetic metal ions $Zn^{2+}(3d^{10})$ and $Mo^{4+}(4d^2)$ in square antiprism geometry, the $\chi T$ values measured in the dark are in agreement with the diamagnetic nature of the compound (Figure 4). Under light irradiation at 405 nm, the value of $\chi T$ increases from 0 to 0.9 cm$^3$ mol$^{-1}$ K after 38 h of irradiation. The reduced magnetizations measured after the light excitation are not superimposed. This is consistent with a weak magnetic anisotropy in the photoinduced state. The saturation value of magnetization at 1.8 K is about 1.15 Nβ. The temperature dependence of the $\chi T$ product from 2 K to 300 K has a similar shape to the $\chi T$ vs. $T$ plot of **1**. For **2**, a maximum value of $\chi T$ product of 0.89 cm$^3$ mol$^{-1}$ K is reached at 15 K. Then, the $\chi T$ product decreases monotonously to reach 0.7 cm$^3$ mol$^{-1}$ K at 240 K. Above this temperature, a faster decrease of the $\chi T$ product is

observed. At 300 K, the $\chi T$ value is equal at 0.3 cm$^3$ mol$^{-1}$ K, well above the value obtained before the light irradiation. The lack of complete reversibility after thermal heating in **2** is quite unusual for a photomagnetic compound containing the [Mo(CN)$_8$]$^{4-}$ unit. This uncommon observation needs further structural investigations to be fully understood. Finally, a new $\chi T$ vs. $T$ plot (red curve on Figure 4) shows that the photo-induced process is not fully erased after a room-temperature treatment. A clear remaining paramagnetic signal around 0.2 cm$^3$ mol$^{-1}$ K is observed.

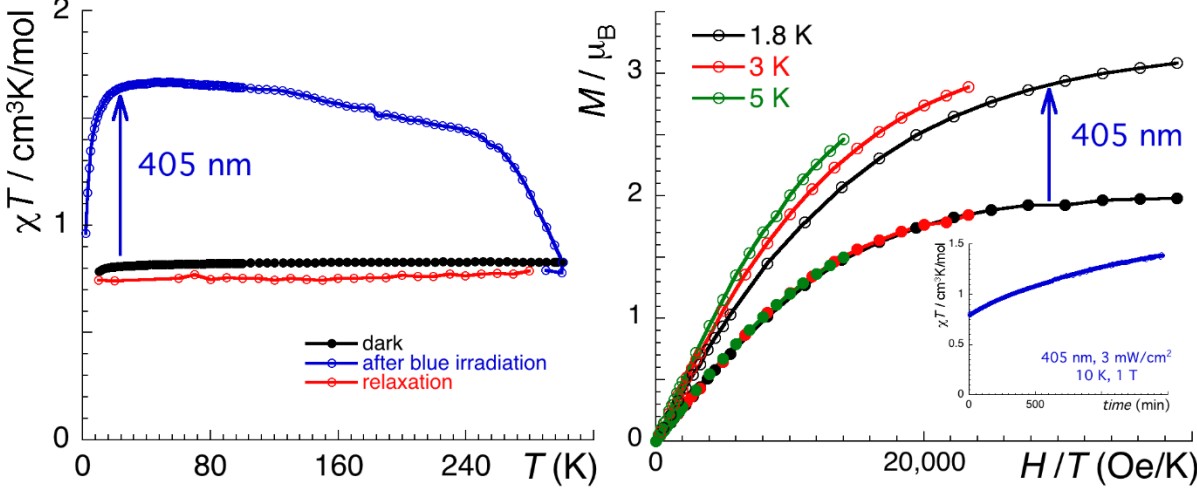

**Figure 3.** (**Left**) $\chi T = f(T)$ plots of **1** measured at 1 T and 0.4 K/min: in the dark before irradiation (dark points), after 405 nm irradiation (open blue points) and after a reconditioning to 300 K (red points). (**Right**) Reduced magnetizations at different temperatures (1.8 K, 3 K and 5 K) before irradiation (full points) and in the photo-excited state of **1** (open points). (Insert: Time dependence of the $\chi T$ for **1** measured at 1 T and 10 K, with continuous irradiation of wavelength at 405 nm (3 mW/cm$^2$)).

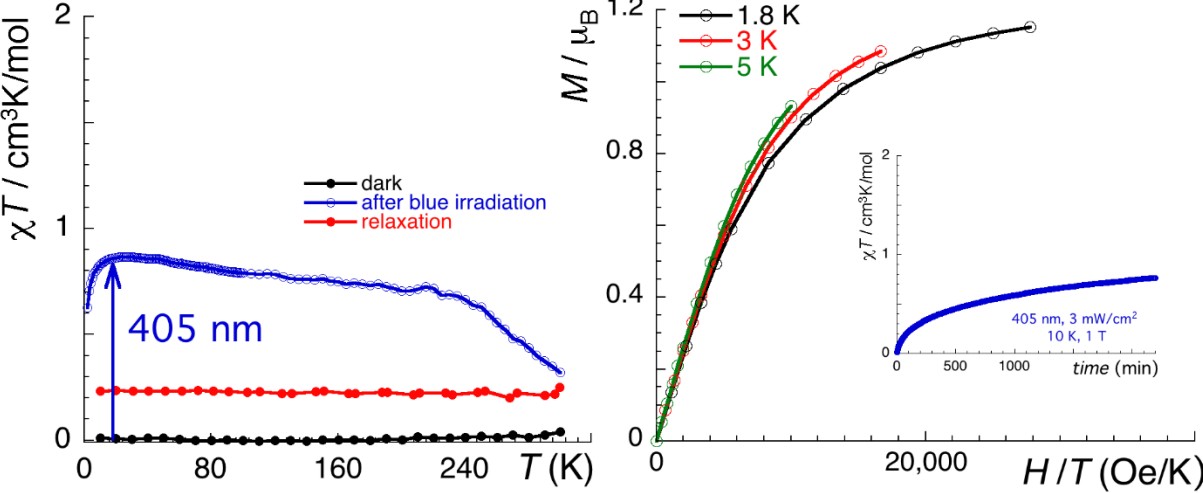

**Figure 4.** (**Left**) $\chi T = f(T)$ plots of **2** measured at 1 T and 0.4 K/min: in the dark before irradiation (dark points), after 405 nm irradiation (open blue points) and after reconditioning at 300 K (red points). (**Right**) Reduced magnetizations at different temperatures (1.8 K, 3 K and 5 K) in the photo-excited state of **2** (open points). (Insert: Time dependence of the $\chi T$ for compound **2** measured at 1 T and 10 K, with continuous irradiation of wavelength of 405 nm (3 mW/cm$^2$)).

### 2.3.3. ZntrenW Compound (**3**)

As evidenced above, **2** and **3** exhibit very similar crystal structures, but they obviously contain a different octacyanometalate anion, albeit in the same geometry. As shown by the magnetic properties of **3** in Figure 5, **3** is a diamagnetic compound in agreement with

the diamagnetic configuration of two $Zn^{2+}(3d^{10})$ ions and the one $W^{4+}(5d^2$ in square antiprism geometry) ion, with $\chi T$ values measured in the dark close to 0. Under light irradiation at 405 nm, the value of $\chi T$ increases from 0 to reach 0.49 cm$^3$ mol$^{-1}$ K after 30 h of irradiation. The reduced magnetizations measured after the light excitation are almost superimposed, consistent with a very weak magnetic anisotropy in the photoinduced state. A clear saturation of magnetization at 1.8 K is observed at the value of 0.51 N$\beta$. The temperature dependence of the $\chi T$ product from 2 K to 300 K is of similar shape to the $\chi T$ vs. $T$ plots of **1** and **2**. For **3**, a plateau is observed with a maximum value of 0.66 cm$^3$ mol$^{-1}$ K around 50 K. Then, a small decrease is observed to reach 0.4 cm$^3$ mol$^{-1}$ K at 200 K. Above this temperature, a faster decrease of the $\chi T$ product is observed, and the $\chi T$ value is back to 0 at 250 K, suggesting that the compound is back in its diamagnetic ground state. This is confirmed by a new $\chi T$ vs. $T$ plot measured after the light excitation and thermal heating of the sample.

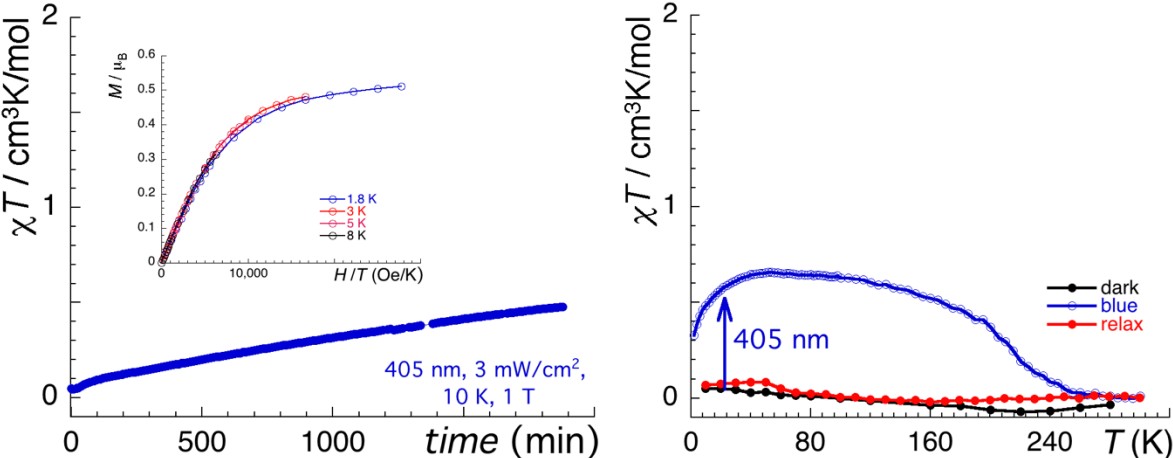

**Figure 5.** (**Left**) Time dependence of the $\chi T$ product measured at 1 T during 405 nm light irradiation (0.3 mW/cm$^2$) (Inset: Reduced magnetizations at different temperatures (1.8 K, 3 K, 5 K and 8 K) in the photo-excited state of **3**). (**Right**) $\chi T$ = f($T$) plots at 1 T and 0.4 K/min of **3** measured in the dark before any light irradiation, after 405 nm irradiation and after reconditioning to 300 K.

## 3. Discussion and Conclusions

In this work, we were able to synthesize three new ionic salts containing the 4d $[Mo(CN)_8]^{4-}$ anion or its 5d analog $[W(CN)_8]^{4-}$ anion. The $[Mo(CN)_8]^{4-}$ anion is known to be involved in several polynuclear compounds exhibiting photomagnetic properties. On the other hand, only few examples of photomagnetic systems based on the $[W(CN)_8]^{4-}$ complex have been reported [20,21]. The three systems reported in this work are ionic salts and are based on large cations of formula of $[\{M'(tren)\}_3(\mu\text{-tren})]^{6+}$ containing 3d metal ions and the $[M(CN)_8]^{4-}$ anions. **1** adopts a slightly different crystal packing than the ones found in **2** and **3** which are almost isostructural if we exclude their solvent content. However, based on the vicinity of the coordination geometries of the metal ions in **1**, **2** and **3**, it is reasonable to compare their respective photomagnetic properties.

The three compounds reported in this work show a significant photomagnetic response. Some of the characteristics of their photo-induced states are common with the other photomagnetic systems based on $[M(CN)_8]^{4-}$ anions. First, the photo-induced states are formed at 10 K with a blue light irradiation. Second, they also have a high thermal stability, and the recovering of the original magnetic properties (i.e., before irradiation) occurs around room temperature. Several mechanisms are proposed in the literature to explain the observed photomagnetic properties: metal-to-metal charge transfer (MMCT) or spin crossover (SCO) mechanisms. The metal-to-metal charge transfer is possible where the $[M(CN)_8]^{4-}$ units can be easily oxidized by the presence of reductive species, as, for example, $Cu^{2+}$ ions. In this case, the presence of a metal-to-metal charge transfer transition (from

$Mo^{4+}$-$Cu^{2+}$ to $Mo^{5+}$-$Cu^{+}$) in optical spectra appearing around 500 nm is the characteristic feature of this mechanism [22]. For compound **1** of this study, no MMCT is observed in its optical spectrum. Another indirect proof of the absence of MMCT mechanism for **1** is the comparison with the photomagnetic properties of **2**. In **2**, the $Cu^{2+}$ ion has been substituted with $Zn^{2+}$ ion which cannot easily form a $Zn^{+}$ ion, thus excluding the MMCT mechanism.

The similarities of photomagnetic properties of **1** and **2** are nicely shown with the photomagnetic difference curves of **1** where the $Cu^{2+}$ contributions are removed by considering the difference of $\chi T$ or $M$ before and after irradiation (Figure 6). The resulting $\chi T$ vs. $T$ and $M$ vs. $H$ plots display a strong similarity with the plots of **2** (Figure 4). This suggests that the photomagnetic properties of **1** and **2** come from the $[Mo(CN)_8]^{4-}$ anions. As mentioned in the introduction, this anion can display a SCO between a $S = 0$ state and a $S = 1$ state. Recently, we have investigated the photo-induced singlet-triplet trapping in the $K_4[Mo(CN)_8]\cdot 2H_2O$ that is accompanied by a breaking of one Mo-CN bond in the crystalline state [6]. To evaluate if this mechanism is active in **1** and **2**, we have analyzed the hypothesis of the photo-induced formation of the triplet state. Because the photomagnetic properties of **2** are not fully reversible, we only analyzed the magnetic data of **1** after the removal of the $Cu^{2+}$ contributions (Figure 6). The triplet state was computed using the anisotropy parameters calculated in [6], namely $|D/k_B| = 20$ K and $g = 1.9$. To simulate correctly the properties of **1**, we also used a partial population of the triplet state at 75% ($p = 0.75$). Figure 6 shows the good reproducibility of the experimental data at a low temperature (T < 120 K), considering the triplet state. At higher temperatures, relaxation that is not considered in the theoretical model probably occurs, and leads to discrepancies with the experiment. This comparative analysis suggests the presence of a SCO mechanism centered on the $[Mo(CN)_8]^{4-}$ anions in **1** and **2**.

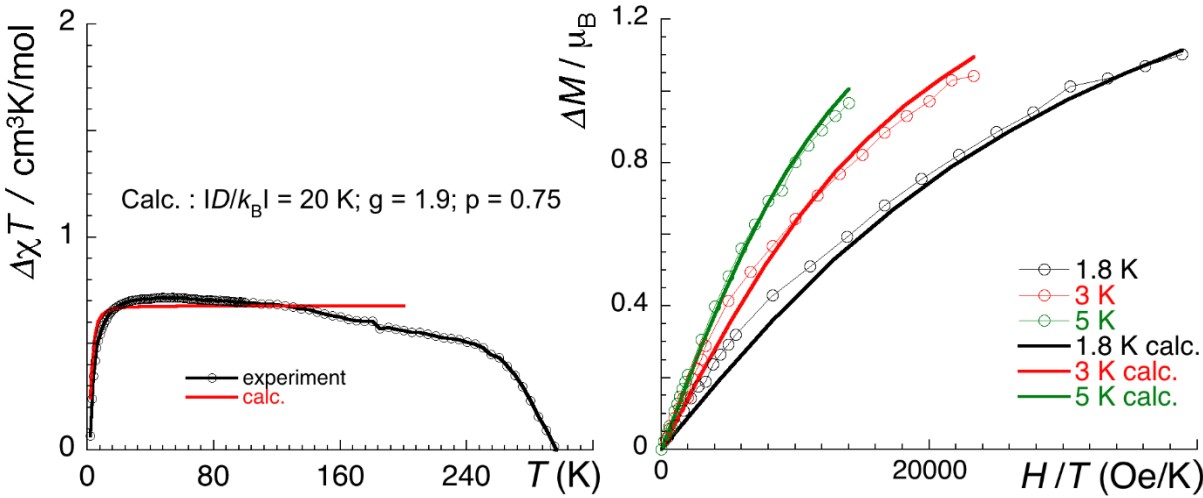

**Figure 6.** (**Left**) $\chi T = f(T)$ plot of **1** obtained with the difference of $\chi T$ before and after irradiation (open dark points). In red, theoretical model for the triplet state (see text). (**Right**) $M = f(H/T)$ of the reduced magnetizations obtained with the difference of $M$ before and after irradiation (open points) at different temperatures. In colored lines, theoretical models for the triplet state (see text).

The photomagnetic properties of the two Zn-based compounds **2** and **3** are similar but **3** displays a lower photoconversion than **2**, as shown by the lower observed values in the $\chi T$ vs. $T$ and $M$ vs. $H$ plots. By analogy with **2**, the observed behavior can be interpreted as a spin crossover from a low spin $S = 0$ to an high spin $S = 1$ for a $5d^2$ complex. Compared to other systems containing the $[W^{IV}(CN)_8]^{4-}$ units, the photoexcited state of **3** has a significant magnetic response, even at 200 K. This feature shows that the photo-induced state in **3** has a lifetime much higher that the lifetimes observed in 3d spin crossover metal ions exhibiting the LIESST phenomenon [23].

To conclude, a series of new photomagnetic compounds based on isolated octa-cyanometalates are reported in this work. Their photomagnetic properties have been analyzed as a photo-induced singlet-triplet crossover on the $[M(CN)_8]^{4-}$. Remarkably, we have shown that the incorporation of $[M^{IV}(CN)_8]^{4-}$ with bulky $[\{M(tren)\}_3(\mu\text{-tren})]^{6+}$ cations leads to a high thermal stability (above 200 K). This is quite an interesting result because the thermal stability of the photo-induced magnetic changes in the reference $K_4[Mo(CN)_8]\cdot2H_2O$ compound is below 65 K [6]. To check if these photo-induced changes are also accompanied with a M-CN bond breaking, as already reported in the $K_4[Mo^{IV}(CN)_8]\cdot2H_2O$ compound, other measurements are necessary such as photocrystal-lography at low temperature. These compounds are not the best candidates for that because of their huge solvent content. Therefore, the solvent composition of the compounds should be improved, for instance, by using organic cations to limit the presence of water during the crystallization and therefore most likely to increase the crystallinity and consequently the accuracy of the crystal structure determination. To further explain the structure and photomagnetic property relationship in ionic compounds built with $[M(CN)_8]^{4-}$ anions, the exploration of other type of cations to change the spatial arrangement of photomagnetic-active $[M^{IV}(CN)_8]^{4-}$ is still highly demanded.

## 4. Materials and Methods

### 4.1. General Remarks

$K_4[Mo^{IV}(CN)_8]\cdot2H_2O$ and $K_4[W^{IV}(CN)_8]\cdot2H_2O$ were synthesized successfully by following the procedures according to the literature [24].

### 4.2. Synthesis

#### 4.2.1. Preparation of CutrenMo (**1**)

Compound **1** was prepared by a layering technique. A mixture of 3 mL solution $CuCl_2\cdot2H_2O$ (35.7 mg, 2.0 mmol) and 528 mg of tren ligand was diffused through 20 mL MeOH:$H_2O$ (1:1) solution into the bottom 1 mL solution of $K_4[Mo^{IV}(CN)_8]\cdot2H_2O$ (100 mg, 2.0 mmol). Green crystals of compound **1** would appear after one week of slow diffusion. Elemental analysis for **1** is as follows. Anal. Calcd for $[[\{Cu(tren)\}_3(\mu\text{-tren})]_{12}[Mo(CN)_8]_6\cdot45H_2O\cdot2MeOH$, Cu12Mo6C146N112H386O47: C, 30.23%; H, 6.71%; N, 27.04%. Found: C, 31.08%; H, 6.29%; N, 27.70% FT-IR (cm$^{-1}$): 3263vs [$\nu$(O-H), $\nu$(N-H)]; 3147m, 2956w, 2923w, 2886w, 2825w [$\nu$(C-H)]; 2091vs [$\nu$(C≡N)]; 1591m [$\gamma$(O-H)]; 1471m, 1311w [$\nu$(C-N), $\nu$(C-C)]; 1101w, 1061ms, 997vs, 981m, 900w, 900w, 872w, 750w, 633s(br) [$\gamma$(N-H out-of-plane)].

#### 4.2.2. Preparation of ZntrenMo (**2**)

In the first step, we mixed a 2 mL solution of $ZnCl_2$ (30.0 mg, 2.2 mmol) with 240 mg solution of tren ligand. The above mixture solution was added slowly to 1 mL solution of $K_4[Mo^{IV}(CN)_8]\cdot2H_2O$ (50 mg, 1.0 mmol) avoid shaking. Then 1.5 mL of MeOH was slowly added, and yellow crystals of compound **2** would appear after one night. FT-IR (cm$^{-1}$): 3257vs [$\nu$(O-H), $\nu$(N-H)]; 3145m, 2964w, 2869w [$\nu$(C-H)]; 2098vs [$\nu$(C≡N)]; 1585m [$\gamma$(O-H)]; 1473m, 1322w [$\nu$(C-N), $\nu$(C-C)]; 1083w, 1054w, 1007ms, 989vs, 885m, 865w, 655s(br) [$\gamma$(N-H out-of-plane)].

#### 4.2.3. Preparation of ZntrenW (**3**)

Similarly to compound **2**, we first mixed a 2 mL solution of $ZnCl_2$ (30.0 mg, 2.2 mmol) with 240 mg solution of tren ligand. The above mixture solution was added slowly to 1 mL solution of $K_4[W^{IV}(CN)_8]\cdot2H_2O$ (58 mg, 1.0 mmol) avoid shaking. Then 1.5 mL of MeOH was slowly added, and yellow crystals of compound **3** would appear after one night. FT-IR (cm$^{-1}$): 3246vs [$\nu$(O-H), $\nu$(N-H)]; 3149m, 2960w, 2917w, 2894w, 2871w [$\nu$(C-H)]; 2091vs [$\nu$(C≡N)]; 1581m [$\gamma$(O-H)]; 1473m, 1322w [$\nu$(C-N), $\nu$(C-C)]; 1083w, 1056w, 1007ms, 983vs, 885m, 865w, 655s(br) [$\gamma$(N-H out-of-plane)].

### 4.3. Physical Measurements

#### 4.3.1. Infrared Spectroscopy

The FT-IR spectra were recorded in the range of 650 $cm^{-1}$–4000 $cm^{-1}$ on a Thermo-Fisher Nicolet$^{TM}$ 6700 ATR (attenuated total reflection) spectrometer equipped with a Smart iTR diamond window on pure solid samples.

#### 4.3.2. UV-Visible Spectroscopy

Solid-state UV-vis-NIR absorption spectra were recorded with a PerkinElmer Lambda 35 UV/vis spectrophotometer equipped with a PerkinElmer Labsphere on pure solid samples.

#### 4.3.3. Magnetic Measurements

All magnetic properties were measured by a Quantum Design MPMS XL system in the range of temperatures of 1.8–300 K. Photomagnetic studies were conducted on a smaller sample (ca. 0.5 mg) sealed in a small PVE bag fixed with Scotch tape, blocked tightly between two transparent polypropylene films and mounted in the probe equipped with an optical fiber entry enabling the transmission of laser light of 405 nm line (P $\approx$ 3 $mW/cm^2$) into the sample space. We used for the three compounds the molecular weights shown in Table A1. To compare the magnetic data of the three compounds, the plots were obtained considering per $M'_2Mo$ or $M'_2W$ units. This means that the molecular weights have been divided by 6 for **1** in Figure 3, and by 3 for **2** and **3** in Figures 4 and 5. Diamagnetism of the sample holders and of the constituent atoms (Pascal's tables) was accounted for in all the obtained magnetic and photomagnetic data.

#### 4.3.4. Powder X-ray Diffraction (PXRD)

PXRD was performed on a PANalytical X'PERT MDP-PRO diffractometer (Cu K$\alpha$ radiation) equipped with a graphite monochromator using the θ-θ Bragg–Brentano geometry. The sample was deposited on a silicon holder for Bragg–Brentano geometry.

#### 4.3.5. Elemental Analysis

Elemental analyses of C, H and N were carried out with a German Elementary Vario EL III instrument.

#### 4.3.6. Single-Crystal X-ray Crystallography

Data collection and reduction for **1** and **2** were performed on a Microfocus rotating anode (Rigaku FRX) operating at 45 kV and 66 mA at the CuK$\alpha$ edge (λ = 1.54184 Å) with a partial chi goniometer. The X-ray source is equipped with high-flux Osmic Varimax mirrors and a Dectris Pilatus 300K detector. Data collection and reduction for **3** were performed on the Bruker Apex II instrument operating at 50 kV and 30 mA using molybdenum radiation Mo K$\alpha$ [λ = 0.71073 Å]. The crystal structures were solved by direct methods using SHELXT and refined using a F2 full-matrix least-squares technique of SHELXL2014/7 [25] included in the OLEX-2 1.2 [26] software packages. The non-H atoms were refined anisotropically, adopting weighted full-matrix least squares on F2. CCDC 2083835, 2083836 and 2083837 contain the supplementary crystallographic data for compounds **1**, **2** and **3**, and additional crystallographic information is available in the Supporting Information. The structural data presented as figures were prepared with the use of the OLEX-2 software. Geometries of metal centers are estimated with the Continuous Shape Measures (CShM) analysis using of SHAPE v2.0 software [27].

**Supplementary Materials:** The following are available online at https://www.mdpi.com/article/10.3390/magnetochemistry7070097/s1, IR, UV-Vis spectra, additional crystallographic tables and figures, additional magnetic figures.

**Author Contributions:** The authors have contributed equally to the conceptualization and execution of this work and wrote the manuscript together. All authors have read and agreed to the published version of the manuscript.

**Funding:** The Chinese Scholarship Council is acknowledged for the phD funding of Xinghui Qi. Financial supports from the CNRS (Pessac), University of Bordeaux and the CPER are acknowledged.

**Institutional Review Board Statement:** Not applicable.

**Informed Consent Statement:** Not applicable.

**Data Availability Statement:** The data are available by corresponding authors.

**Acknowledgments:** We are grateful for the fruitful discussions with S. Pillet and E. E. Bendeif in Lorraine University.

**Conflicts of Interest:** The authors declare no conflict of interest.

## Appendix A

**Table A1.** Crystal and experimental data for **1**, **2** and **3**.

| Compound/CCDC Number | 1/2083835 | 2/2083836 | 3/2083837 |
|---|---|---|---|
| Formula | $Mo_6Cu_{12}C_{146}H_{386}N_{112}O_{47}$ | $Mo_3Zn_6C_{72}H_{180}N_{56}O_{18}$ | $W_3Zn_6C_{72}H_{178}N_{56}O_{17}$ |
| $D_{calc.}$/g cm$^{-3}$ | 1.424 | 1.406 | 1.537 |
| $\mu$mm$^{-1}$ | 4.121 | 4.036 | 3.760 |
| Formula Weight | 5801.61 | 2798.68 | 3044.39 |
| $T$/K | 130(2) | 130(2) | 150(2) |
| Crystal System | monoclinic | monoclinic | monoclinic |
| Space Group | $P2_1/c$ | $Cc$ | $Cc$ |
| $a$/Å | 30.4150(4) | 32.1487(9) | 31.8599(9) |
| $b$/Å | 37.1820(5) | 16.8699(3) | 16.9574(3) |
| $c$/Å | 22.5401(2) | 25.4013(6) | 25.3107(6) |
| $\beta$/° | 98.0410(10) | 106.487(3) | 106.013(3) |
| V/Å$^3$ | 25239.8(5) | 13209.8(6) | 13143.8(6) |
| Z | 4 | 4 | 4 |
| Z′ | 1 | 1 | 1 |
| Wavelength/Å | 1.54184 | 1.54184 | 0.71073 |
| Radiation type | CuK$_a$ | CuK$_a$ | MoK$_a$ |
| $Q_{min}$/° | 2.309 | 2.867 | 1.701 |
| $Q_{max}$/° | 73.591 | 74.083 | 26.733 |
| Measured refl. | 199397 | 46339 | 151852 |
| Measured indep. refl. | 49550 | 19947 | 27860 |
| Observed indep. refl. | 35018 | 19214 | 22758 |
| $R_{int}$ | 0.0760 | 0.0775 | 0.0796 |
| Parameters | 2908 | 1450 | 1438 |
| Restraints | 0 | 2 | 20 |
| Largest diff. peak (e-.A$^{-3}$) | 19.876 | 1.7437 | 1.938 |
| Deepest diff. hole (e-.A$^{-3}$) | −2.959 | −1.561 | −1.187 |
| GooF (S) | 1.764 | 1.042 | 1.018 |
| $wR_2$ (all data) | 0.4624 | 0.2071 | 0.0959 |
| $wR_2$ | 0.4298 | 0.2049 | 0.0894 |
| $R_1$ (all data) | 0.2056 | 0.0791 | 0.0574 |
| $R_1$ | 0.1830 | 0.0769 | 0.0397 |

**Table A2.** Continuous shape measurements (CShM) for metal ions in **1**, **2** and **3**.

| [{Cu(tren)}$_3$(μ-tren)]$_{12}$[Mo(CN)$_8$]$_6$·45H$_2$O·2MeOH **1** | | | | | | |
|---|---|---|---|---|---|---|
| Mo(CN)$_8$ | Mo1 | Mo2 | Mo3 | Mo4 | Mo5 | Mo6 |
| *SAPR* | **0.218** | **0.254** | **0.535** | **0.230** | **0.652** | **0.321** |
| *TDD* | 2.201 | 2.008 | 1.679 | 1.899 | 1.232 | 1.794 |
| *JBTPR* | 2.821 | 2.284 | 2.252 | 2.660 | 1.971 | 2.360 |
| *BTPR* | 2.220 | 1.709 | 1.649 | 2.077 | 1.384 | 1.738 |

**Table A2.** *Cont.*

| [{Cu(tren)}$_3$(μ-tren)]$_{12}$[Mo(CN)$_8$]$_6$·45H$_2$O·2MeOH **1** | | | | | | |
|---|---|---|---|---|---|---|
| **CuN$_5$** | **Cu1** | **Cu2** | **Cu3** | **Cu4** | **Cu5** | **Cu6** |
| *TBPY* | **0.420** | **0.377** | **0.904** | **0.540** | **0.368** | **0.443** |
| *SPY* | 4.269 | 4.425 | 3.226 | 4.175 | 4.805 | 4.970 |
| *JTBPY* | 3.967 | 3.731 | 3.928 | 3.661 | 3.794 | 3.911 |
| **CuN$_5$** | **Cu7** | **Cu8** | **Cu9** | **Cu10** | **Cu11** | **Cu12** |
| *TBPY* | **0.380** | **0.321** | 2.225 | **0.348** | **0.514** | **0.789** |
| *SPY* | 4.975 | 4.832 | **1.491** | 5.009 | 4.016 | 3.401 |
| *JTBPY* | 3.966 | 3.639 | 4.730 | 3.705 | 4.016 | 3.977 |
| [{Zn(tren)}$_3$(μ-tren)]$_2$[Mo(CN)$_8$]$_3$·18H$_2$O **2** | | | | | | |
| **Mo(CN)$_8$** | **Mo1** | **Mo2** | **Mo3** | | | |
| *SAPR* | **0.232** | **0.590** | **0.836** | | | |
| *TDD* | 2.106 | 1.567 | 1.372 | | | |
| *JBTPR* | 2.245 | 1.907 | 1.819 | | | |
| *BTPR* | 1.663 | 1.422 | 1.111 | | | |
| **ZnN$_5$** | **Zn1** | **Zn2** | **Zn3** | **Zn4** | **Zn5** | **Zn6** |
| *TBPY* | **0.817** | **0.773** | **1.076** | **0.771** | **0.814** | **0.760** |
| *SPY* | 5.582 | 4.987 | 5.149 | 5.715 | 4.698 | 5.558 |
| *JTBPY* | 2.594 | 2.368 | 2.152 | 2.654 | 2.510 | 2.548 |
| [{Zn(tren)}$_3$(μ-tren)]$_2$[W(CN)$_8$]$_3$·17H$_2$O **3** | | | | | | |
| **W(CN)$_8$** | **W1** | **W2** | **W3** | | | |
| *SAPR* | **0.969** | **0.295** | **0.532** | | | |
| *TDD* | 1.110 | 1.922 | 1.781 | | | |
| *JBTPR* | 1.868 | 2.104 | 1.934 | | | |
| *BTPR* | 1.151 | 1.540 | 1.355 | | | |
| **ZnN$_5$** | **Zn1** | **Zn2** | **Zn3** | **Zn4** | **Zn5** | **Zn6** |
| *TBPY* | **0.792** | **0.916** | **0.742** | **0.976** | **0.852** | **0.889** |
| *SPY* | 5.393 | 4.487 | 5.890 | 4.707 | 4.803 | 5.930 |
| *JTBPY* | 2.514 | 2.535 | 2.449 | 2.285 | 2.290 | 2.413 |

The numbers in the tables correspond to the S shape measures relative to the square antiprism (SAPR), triangular dodecahedron (TDD J84), Johnson elongated triangular bipyramid (JBTPR J14) and biaugmented trigonal prism (BTPR J50) for M(CN)$_8$ unit; Trigonal bipyramid (TBPY), Spherical square pyramid (SPY) and Johnson trigonal bipyramid (JTBPY J12) for M'N$_5$ unit. When the respective shape measure parameter equals zero, the real geometry coincides with the idealized one. For each site, the minimum calculated shape measure is given in violet.

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
