# Peer review of "New Photomagnetic Ionic Salts Based on [MoIV(CN)8]4− and [WIV(CN)8]4− Anions†"

_magnetochemistry, doi:10.3390/magnetochemistry7070097_

Round 1
Reviewer 1 Report
The paper describes the syntheses and the characterization of three complexes presenting photomagnetic effect.
The paper is very well written and the photo effect on this type of complexes is well stablished in related previous compounds and the present paper provides a nice introduction about the topic.
There are some typos in the text, but after the correction I would recommend the publication in Magnetochemistry without changes.
Author Response
Comments from referee 1: “The paper describes the syntheses and the characterization of three complexes presenting photomagnetic effect.
The paper is very well written and the photo effect on this type of complexes is well stablished in related previous compounds and the present paper provides a nice introduction about the topic.
There are some typos in the text, but after the correction I would recommend the publication in Magnetochemistry without changes.”
Our response. We are very happy to read that the referee appreciates our work. We have corrected some typos in the text as suggested by the referee. These corrections are underlined in yellow in our revised manuscript.

Reviewer 2 Report
The manuscript describes a study of new photomagnetic complexes based on M(CN)84– anions with large coordination cations. All complexes are well characterized by different physicochemical methods. It was shown, that the photo-induced effect in the complexes is accompanied by a spin change, and separation of the anionic moieties in the crystals lead to the high thermal stability of the excited state up to 240-250 K.
The manuscript is well written, but minor fixes are needed:
1) Some important details should be added to the manuscript:
– I suppose that the FT-IR spectra were recorded for solids, like to UV−vis− NIR absorption spectra, not for KBr pellets.
– The magnetic field strength for χT(T) dependencies should be indicated.
–Accordingly to the SCXRD data, there are a lot of solvent molecules in the structure. So, how the molar paramagnetic susceptibilities were calculated? What molecular weight values were used? Is it possible to estimate solv content by the elemental analysis?
– In the Physical Measurements part an information about PXRD method is absent.
2) In the abstract the second and third sentences are the same. One of them is excessive.
3) The manuscript should be checked for misprints (e.g. lines 20 – importanta, 23 – The obtainedse photomagnetic, 275 – synthetize, ref.2 – and reltaed compounds, in the Figure S3 some W-C-N fragments are non-linear, check the angle values in the Table S6).

Author Response
Comments from referee 2: “The manuscript describes a study of new photomagnetic complexes based on M(CN)84–anions with large coordination cations. All complexes are well characterized by different physicochemical methods. It was shown, that the photo-induced effect in the complexes is accompanied by a spin change, and separation of the anionic moieties in the crystals lead to the high thermal stability of the excited state up to 240-250 K.“
Our response: We warmly thank the reviewer for the positive feedback and pointing out non-negligible points to help improve the manuscript.
The manuscript is well written, but minor fixes are needed:
1) Some important details should be added to the manuscript:
– I suppose that the FT-IR spectra were recorded for solids, like to UV−vis− NIR absorption spectra, not for KBr pellets.
Our response: Yes. The FT-IR spectra are recorded with pure compounds without KBr dispersion. We tried to clarify ourselves by adding “for pure solid sample” in the Physical Measurements part.
– The magnetic field strength for χT(T) dependencies should be indicated.
Our response: OK. We have added “measured at 1 T” for the corresponding figures.
–Accordingly to the SCXRD data, there are a lot of solvent molecules in the structure. So, how the molar paramagnetic susceptibilities were calculated? What molecular weight values were used? Is it possible to estimate solv content by the elemental analysis?
Our response: We do have plenty of solvent molecules in the structures of the three compounds. We have estimated the number of water and methanol solvent molecules from the CIF files. But after a careful reading on the manuscript, we have realized that some mistakes have been introduced in the table. In the revised version, we have corrected the Table A1 with the correct values. To avoid confusion for the readers, we have decided to fix the number of solvents molecules for the three compounds (see below). All the formulas have been modified in the revised manuscript. We have added the following sentence in the Physical Measurements part “We used the molecular weights considering 45 H2O and 2 MeOH molecules for compound 1, 18 H2O molecules for compound 2 and 17 H2O molecules for compound 3”.
We have also added the elemental analysis (EA) data in the Syntheses part for compound 1: “Elemental analysis for 1 is as follows. Anal. Calcd for [[{Cu(tren)}3(μ-tren)]4[Mo(CN)8]6·45H2O·2MeOH, Cu12Mo6C146N112H386O47: C, 30.23%; H, 6.71%; N, 27.04%. Found: C, 31.08%; H, 6.29%; N, 27.70%.”
There is a slight deviation between the number of solvents determined by X-ray structures and EA. Considering the high number of solvent molecules in our compounds, it is reasonable to think that the solvent content can be different from one crystal to another one. On the opposite, EA gives average numbers on microcrystalline powders. Unfortunately, we were not able to have EA for compounds 2 and 3 for technical issues. To be consistent for the three compounds, we have decided to use the molecular weights determined by crystal structures in the paper. Moreover, if we consider the EA results, the best agreements with the experimental % will give the final formula for 1: Cu12Mo6C144N112H368O40 meaning that 1 contains only 40 water molecules (see table).
|
Mol. Weight A Cu12Mo6C146N112H386O47 1·45H2O·2CH3OH (From CIF) |
Mol. Weight B Cu12Mo6C144N112H368O40 1·40H2O (From EA) |
Experimental |
C |
30.23% |
30.63% |
31.08% |
N |
27.04% |
27.78% |
27.70% |
H |
6.71% |
6.57% |
6.29% |
To be sure that the conclusions of the paper are not modified using the molecular weights determined by crystal structures, we have treated the magnetic properties using the 2 molecular weights (see table above). As the photomagnetic properties are measured with a small quantity of compound (< 0.5 mg), the diamagnetic correction from the sample holder and the bag where the sample is inserted is important. Changing the molecular weight implies to modify the diamagnetic corrections. Consequently, the values of susceptibilities are very similar.
To conclude, we have added in the revised version of the manuscript the two following modifications: “Elemental analysis for 1 is as follows. Anal. Calcd for [[{Cu(tren)}3(μ-tren)]12[Mo(CN)8]6·45H2O·2MeOH, Cu12Mo6C146N112H386O47: C, 30.23%; H, 6.71%; N, 27.04%. Found: C, 31.08%; H, 6.29%; N, 27.70%.”
“To calculate the magnetic susceptibilities, and to be able to compare the compounds, the figures 3, 4 and 5 have been plotted considering the magnetism per M’2Mo or M’2W units. This means that the molecular weights have been divided by 6 for 1 in Figure 3, and by 3 for 2 and 3 in Figures 4 and 5.” The last sentence has been added in 4.3.3 “Magnetic measurements” section.
– In the Physical Measurements part an information about PXRD method is absent
Our response: We added in the Physical Measurements part:
4.3.4. Powder X-ray diffraction (PXRD) analysis the following sentence:
“PXRD was performed on a PANalytical X’PERT MDP-PRO diffractometer (Cu-Kα radiation) equipped with a graphite monochromator using the θ-θ Bragg-Brentano geometry. The sample was deposited on a silicon holder for Bragg-Brentano geometry.
4.3.5. Elemental analysis
Elemental analyses of C, H, and N were carried out with a German Elementary Vario EL III instrument.”
2) In the abstract the second and third sentences are the same. One of them is excessive.
Our response: We have corrected the abstract in the revised version.
3) The manuscript should be checked for misprints (e.g. lines 20 –importanta, 23 –The obtainedse photomagnetic, 275 –synthetize, ref.2 –and reltaed compounds, in the Figure S3 some W-C-N fragments are non-linear, check the angle values in the Table S6).
Our response: We are grateful for the reviewer for pointing out those typos and we are sorry for the inconvenience caused by this. Because we went further in the refinement of the structure of compound 3, the Figure S3 was not obtained from our final version of CIF. We have redrawn the Figure S3. We thank for the reviewer to pointing out this mistake.
Reviewer 3 Report
Mathoniere and co-authors present a nice set of data, including synthesis and crystallographic characterization of three [M(CN)8] coordination salts, and report their magnetic and photomagnetic properties. The work is well performed. My congratulations! However, some minor issues should be addressed before publishing.
The exact composition of investigated materials is not clear to me. Although I do not have access to CIF files, I expect that at least the number of water and methanol solvent molecules can be identified from X-ray structures. Indeed, in the main text, the amount of solvent molecules is not specified, however, formulas and molecular weights of complexes are provided in Table A1. The knowledge about the (large) amount of solvent molecules is then necessary to perform diamagnetic corrections properly… Furthermore, I was surprised not to find elemental analysis data, which can further help to estimate the amount of solvent molecules in investigated materials.
The photomagnetic effect on 2 is not fully reversible. The authors should comment on it.
Line 296. Where an MMCT band is expected? References are needed. The position of MMCT band should be dependent on M…M distance and/or interconnecting ligand(s). Can the authors provide a proper system for comparison?
Can the enhanced thermal stability of the photo-induced state be related to cation-anion interactions observed in this case?
One more reference in the introduction might be included: photoinduced coordination/decoordination in bistable cobalt complexes is known: DOI: 10.1039/C5SC00130G
Some typos:
importanta
obtainedse
Author Response
Comments from referee 3: “Mathoniere and co-authors present a nice set of data, including synthesis and crystallographic characterization of three [M(CN)8] coordination salts, and report their magnetic and photomagnetic properties. The work is well performed. My congratulations! However, some minor issues should be addressed before publishing.”
Our response: We thank the reviewer for the positive feedback.
The exact composition of investigated materials is not clear to me. Although I do not have access to CIF files, I expect that at least the number of water and methanol solvent molecules can be identified from X-ray structures. Indeed, in the main text, the amount of solvent molecules is not specified, however, formulas and molecular weights of complexes are provided in Table A1. The knowledge about the (large) amount of solvent molecules is then necessary to perform diamagnetic corrections properly… Furthermore, I was surprised not to find elemental analysis data, which can further help to estimate the amount of solvent molecules in investigated materials.
Our response: The CIFs files are accessible on the Cambridge Data Base using the numbers given in table A1, ie. 2083835 For 1, 2083836 for 2 and 2083837 for 3. We do have plenty of solvent molecules in the structures of the three compounds. We have estimated the number of water and methanol solvent molecules from the CIF files. But after a careful reading on the manuscript, we have realized that some mistakes have been introduced in the table. In the revised version, we have corrected the Table A1 with the correct values. To avoid confusion for the readers, we have decided to fix the number of solvents molecules for the three compounds (see below). All the formulas have been modified in the revised manuscript. We have added the following sentence in the Physical Measurements part “We used for the three compounds the molecular weights shown in Table A1. To compare the magnetic data of the three compounds, the plots have been obtained considering per M’2Mo or M’2W units. This means that the molecular weights have been divided by 6 for 1 in Figure 3, and by 3 for 2 and 3 in Figures 4 and 5.”.
We have also added the elemental analysis (EA) data in the Synthesis part for compound 1: “Elemental analysis for 1 is as follows. Anal. Calcd for [[{Cu(tren)}3(μ-tren)]4[Mo(CN)8]6·45H2O·2MeOH, Cu12Mo6C146N112H386O47: C, 30.23%; H, 6.71%; N, 27.04%. Found: C, 31.08%; H, 6.29%; N, 27.70%.”
There is a slight deviation between the number of solvents determined by X-ray structures and EA. Considering the high number of solvent molecules in our compounds, it is reasonable to think that the solvent content can be different from one crystal to another one. On the opposite, EA gives average numbers on microcrystalline powders. Unfortunately, we were not able to have EA for compounds 2 and 3 for technical issues. To be consistent for the three compounds, we have decided to use the molecular weights determined by crystal structures in the paper. Moreover, if we consider the EA results, the best agreements with the experimental % will give the final formula for 1: Cu12Mo6C144N112H368O40 meaning that 1 contains only 40 water molecules (see table).
|
Mol. Weight A Cu12Mo6C146N112H386O47 1·45H2O·2CH3OH (From CIF) |
Mol. Weight B Cu12Mo6C144N112H368O40 1·40H2O (From EA) |
Experimental |
C |
30.23% |
30.63% |
31.08% |
N |
27.04% |
27.78% |
27.70% |
H |
6.71% |
6.57% |
6.29% |
To be sure that the conclusions of the paper are not modified using the molecular weights determined by crystal structures, we have treated the magnetic properties using the 2 molecular weights (see table above). As the photomagnetic properties are measured with a small quantity of compound (< 0.5 mg), the diamagnetic correction from the sample holder and the bag where the sample is inserted is important. Changing the molecular weight implies to modify the diamagnetic corrections. The consequence is that the values of susceptibilities are very similar.
To conclude, we have added in the revised version of the manuscript the following modification: “Elemental analysis for 1 is as follows. Anal. Calcd for [[{Cu(tren)}3(μ-tren)]12[Mo(CN)8]6·45H2O·2MeOH, Cu12Mo6C146N112H386O47: C, 30.23%; H, 6.71%; N, 27.04%. Found: C, 31.08%; H, 6.29%; N, 27.70%.”
The photomagnetic effect on 2 is not fully reversible. The authors should comment on it.
Our response: The photomagnetic properties of the 3 compounds have been investigated at least 2 times for 2 different batches. Only the compound 2 shows a partial reversibility. This peculiar behavior is maybe caused by the fact that the full relaxation occurs above room temperature. To test the hypothesis, we have performed a photomagnetic study with a thermal heating up to 320 K after light irradiation. But the same partial reversibility is still observed with a remaining cT of 0.2 cm3 K/mol. A hypothesis to explain this lack of complete reversibility is that the compound evolves during the thermal heating after the light irradiation. Actually, the light irradiation induces a Mo-CN breaking as observed in the reference compound K4Mo(CN)8·2H2O. This structural reorganization involves an important move of the CN ligand that may create structural differences with the initial state. To prove that, photocrystallographic studies are necessary. But the quality of the structures for the compounds under study is not suitable to perform such studies. In the revised manuscript, we add the following sentence.
“The lack of complete reversibility after thermal heating in 2 is quite unusual for a photomagnetic compound containing the [Mo(CN)8]4-unit. This uncommon observation needs further structural investigations to be fully understood.”
Line 296. Where an MMCT band is expected? References are needed. The position of MMCT band should be dependent on M…M distance and/or interconnecting ligand(s). Can the authors provide a proper system for comparison?
Our response: All the known Cu/Mo(CN)8 compounds which present in their optical spectra a metal-to-metal charge transfer band are around 510 nm corresponding to the conversion of CuII-MoIV pairs into CuI-MoV pairs. The reference 22 (Ohkoshi, S. I. et al J. Am. Chem. Soc. 2006, 128, 270-277) has been added as an example in the revised manuscript.
Can the enhanced thermal stability of the photo-induced state be related to cation-anion interactions observed in this case?
Our response: As the reviewer noticed, we suspect that the increasing size of the cations help in the trapping of the dissociated CN from the Mo or W ions. To check this hypothesis, it will be interesting to perform photo-crystallographic studies at low temperature. But the quality of the structures for the compounds in this study is not enough to perform such studies. We are now studying a new family of complexes to understand this specific point.
One more reference in the introduction might be included: photoinduced coordination/decoordination in bistable cobalt complexes is known: DOI: 10.1039/C5SC00130G
Our response: The reference mentioned by the referee does not correspond to coordination/decoordination process. In the revised mansucript we add in the introduction the following sentence because we realised that we forget an important reference from Sato group with a coordination/decoordination process.
“One example is based on a Co complexe [11] that exhibits a dynamic bond switching with a modulation of the ligand field and the orbital momentum of the metal ion. The second example is shown in a family of on spin crossover FeII [12,13] complexes where the decoordination/coordination process can thermally and photo-induced, as it is the case in the K4[MoCN8]·2H2O complex [6].”
Some typos:
importanta
obtainedse
Our response: We are grateful for the reviewer for pointing out those typos and we are sorry for the inconvenience caused by this.